# Getting Connected to M-Health Technologies through a Meta-Analysis

**DOI:** 10.3390/ijerph20054369

**Published:** 2023-02-28

**Authors:** Luiz Philipi Calegari, Guilherme Luz Tortorella, Diego Castro Fettermann

**Affiliations:** 1Department of Industrial Engineering, Federal University of Santa Catarina, Florianópolis 8040-900, SC, Brazil; 2Department of Mechanical Engineering, University of Melbourne, Melbourne, VIC 3010, Australia

**Keywords:** m-health, information technology, Internet of Things, consumer acceptance, consumer behavior, healthcare, health management, mobile health, management technology, meta-regression

## Abstract

The demand for mobile e-health technologies (m-health) continues with constant growth, stimulating the technological advancement of such devices. However, the customer needs to perceive the utility of these devices to incorporate them into their daily lives. Hence, this study aims to identify users’ perceptions regarding the acceptance of m-health technologies based on a synthesis of meta-analysis studies on the subject in the literature. Using the relations and constructs proposed in the UTAUT2 (Unified Theory of Acceptance and Use of Technology 2) technology acceptance model, the methodological approach utilized a meta-analysis to raise the effect of the main factors on the Behavioral Intention to Use m-health technologies. Furthermore, the model proposed also estimated the moderation effect of gender, age, and timeline variables on the UTAUT2 relations. In total, the meta-analysis utilized 84 different articles, which presented 376 estimations based on a sample of 31,609 respondents. The results indicate an overall compilation of the relations, as well as the primary factors and moderating variables that determine users’ acceptance of the studied m-health systems.

## 1. Introduction

The literature indicates an increase in the concern with people’s health, resulting in a rise in the development of technological products [1]. The continuous development of technologies applied to healthcare has provided patients with a better quality of life while increasing their expectancy of better treatments in the health system [2,3,4]. As a way to improve healthcare, new technologies have started to be incorporated into healthcare systems, such as e-health technologies.

E-health technologies are considered an emerging and growing field in the medical sector [5,6]. The evolution of the development of e-health technologies presents promising alternatives for healthcare carried out effectively and at a low cost [7]. Faced with the growing concern of people with their health, the development of e-health technologies for the remote monitoring of users has presented a significant market evolution [8,9,10]. Technological advances in Internet of Things (IoT) devices, big data strategies, and portable biosensors have generated alternatives to provide personalized e-health services [11]. The greater flexibility in the use of IoT devices, such as wearables, provided by the evolution of cloud computing technologies, promotes the expansion of the use of mobile devices aimed at health services, called m(mobile)-health [12,13]. M-health technologies propose providing health services anytime and anywhere, overcoming temporal and geographic barriers [14]. Highlighting m-health technologies, the demand for wearable devices continues in constant growth [15]. For the wearables market, an annual growth rate of 20% is estimated for the following years, moving about 150 billion euros up to 2028 [16]. The proliferation of wearables on the market predicted for the next decade will stimulate the technological advancement of such devices, improving intelligent systems and their resources [17].

Despite the benefits of using e-health technologies, there needs to be more understanding of the relations among suppliers, technologies, and potential consumers [18]. Furthermore, the rollout of e-health devices must consider all factors that affect the utility perceived by consumers [18]. As a way to understand the consumer’s acceptance, it is essential to interpret the factors that explain the acceptance of new technologies by potential users [19,20,21,22,23]. In the context of studies with models of technological acceptance, studies with small samples hardly consolidate general trends in terms of acceptance [24]. In the particular case of m-health technologies, the literature reports several studies with divergent estimations of m-health acceptance [25,26,27]. A case of this divergence is the relation of the effort expectation construct to the Behavioral Intention to Use construct, reported as having a significant and positive effect [26,28] and in other cases reported as having no significant effect [29,30]. The same case has been reported in the relation between the Facilitating Conditions and the Behavioral Intention to Use constructs, which is significant in some studies [27,31] and not significant in others [32,33]. The frequent divergences in the estimations of the acceptance of m-health technologies in the literature raise the need to identify a general trend among various estimations carried out in particular contexts. In order to deal with the variety of estimations, the literature suggests applying the meta-analysis methodology, which establishes a robust research model based on gathering studies from a specific area [34]. The meta-analysis also raises general trends between divergent results and makes evident the consensus among similar relations [35].

This study aims to identify users’ perceptions regarding the acceptance of m-health technologies based on a synthesis of meta-analysis studies on the subject in the literature. A meta-analysis was carried out using the relations and constructs proposed in the Unified Theory of Acceptance and Use of Technology 2 (UTAUT2) model of technology acceptance proposed by Venkatesh et al. [36] and widely utilized in the m-health literature [37,38,39]. Moreover, the moderation effect of variables proposed in the UTAUT2 was estimated using a meta-regression [34,40] procedure. The results indicate an overall compilation of the relations, as well as the primary factors and moderating variables that determine the users’ acceptance of m-health technologies.

## 2. Acceptance Models

IoT adoption promotes many benefits for industry, companies, and users [21]. However, it is possible to observe in the literature various barriers related to the lack of acceptance of these new technologies by their potential users [41,42,43,44,45]. The divergence among the technology acceptance estimations reported in the literature undermines the reliability of these results [46]. The diversity in the results could be associated with the use of small samples as well as the sampling procedures utilized [6], and meta-analysis is the technique suggested to deal with it and raise robust and reliable estimations [47].

Technology acceptance models have been widely applied to understand user behavior toward various solutions. For example, we have its application in studies on applications and information systems for agricultural activities [48,49], virtual reality systems [50], home devices [24,51,52], autonomous cars, [53], safety systems for construction workers [54], learning environments or e-learning [55,56], e-shopping [57], e-services [58], digital content marketing for tourism [59], mobile payments [60], the visual design of wearables [61], and wearable locating systems [62], among others. The number of new e-health technologies has increased the use of technology acceptance models to improve the comprehension of the factors that affect the user’s acceptance of m-health technologies [63,64,65].

The literature reports various approaches for measuring the acceptance and use of new technologies, such as the technology acceptance model (TAM) [66], the Theory of Planned Behavior (TPB) [67], the Theory of Reasoned Action (TRA) [68], and the Unified Theory of Acceptance and Use of Technology (UTAUT) [69]. Despite the various alternatives, the TAM is one of the most disseminated in the literature [70,71,72,73,74]. However, the TAM model also is criticized for providing an overly generic estimate of user perception relative to the acceptance of new technologies [75].

The UTAUT model was developed as a result of the TAM model’s limitations, presenting a broad application in the literature to measure the acceptance of new e-health technologies [76,77,78]. The UTAUT model proposed by Venkatesh et al. [69] is formed by the following constructs: Performance Expectancy (PE, also expressed as Perceived Usefulness, Extrinsic Motivation, Job-fit, Relative Advantage, and Outcome Expectation), defined as the extent to which a person believes that using a specific system will improve their performance in carrying out a specific action [66]; Effort Expectancy (EE, also expressed as Perceived Ease of Use, Complexity, and Ease of Use), defined as the extent to which a person believes that using a given system will be effortless [66,79]; Social Influence (SI, also expressed as Subjective Norm, Social Factors, and Image), defined as the extent to which an individual believes that people of reference may influence the use of a given system [69]; Facilitating Conditions (FC, also expressed as Perceived Behavioral Control and Compatibility), defined as the extent to which an individual believes in the existence of technical and organizational infrastructure and favorable environmental conditions that motivate them to use technological systems [69]; Behavioral Intention to Use (BI), defined as the extent to which an individual formulates a conscious plan to execute or not execute a future behavior [8,80]; and use behavior (UB), defined as the Usage Behavior measured from the actual frequency of use of a given technology [69].

In order to improve the UTAUT model prediction, the authors proposed including factors related to the consumer’s context, creating the UTAUT2 model [36]. The update brought three new constructs: Hedonic Motivation (HM), defined as the pleasure or fun derived from using the new technology [81,82]; Price Value (PV), which refers to the exchange that the consumer deems fair between the perceived benefits and the monetary costs [83,84]; and Habit, which refers to a reflexive behavior by people or automatic behaviors stemming from their experiences and learning [82,85,86]. Some authors also use UAUT2 variations, adding the Attitude construct (AT) [87,88], which refers to the degree to which the person has a behavior favorable to the use of the technology studied [67].

## 3. Materials and Methods

### 3.1. Proposed Model

Due to the context of m-health technologies, the literature recommends including other constructs in the UTAUT2 model [89]. Hence, besides the relations proposed by the UTAUT2 model, the literature also suggests five other relations of constructs considered critical to the acceptance of m-health technologies. The first relation included is between the Effort Expectancy (EE) > Performance Expectancy (PE) constructs, as suggested by the literature [90,91]. The second relation added is between Performance Expectancy (PE) > Attitude (AT) [92,93]. The third relation suggested is between Effort Expectancy (EE) > Attitude (AT) [94,95]. The fourth relation frequently estimated in the literature is between the constructs Attitude 138 (AT) > Behavioral Intention (BI) [96,97]. The fifth relation added is between Privacy Risks (RP) > Behavioral Intention (BI), also often estimated in the m-health literature [90,98].

The literature suggests that the perceived utility is more significant insofar as wearable technologies are easy to use, i.e., require less user effort [99,100]. Hence, it becomes important to consider the positive effect on the Effort Expectancy (EE) > Performance Expectancy (PE), the sixth relation in the proposed model. The Performance Expectancy (PE) and the Effort Expectancy (EE) are constructs within the cognitive scope that affect the Attitude (AT) of users and, subsequently, determine their intention to use [91]. Attitude is defined as an affective reaction by an individual when using a technology [69]. The literature suggests that more positive attitudes by an individual toward a technology tend to positively influence the Behavioral Intention to Use this technology [91,92]. Hence, the seventh relation, PE > AT, the eighth relation, EE > AT, and the ninth relation, AT > BI, will also be analyzed in this study. Lastly, the increase in the frequency of health data sharing in cloud computing environments stimulates the concern of digital media users with the privacy and security of personal information [98]. The behavior of users avoiding using digital media that require access to personal data becomes one of the most common reasons for not accepting a technology [101]. Hence, it is essential to observe if there is a negative PR > BI effect.

#### 3.1.1. Moderators

Previous studies have pointed out that social characteristics significantly impact users’ acceptance of new technologies and must be incorporated into the acceptance models [102,103]. The current meta-analysis includes the moderator effect on the relationships of two user variables, gender and age. Moreover, the meta-analysis also estimates the moderator effect of time on the relationships. The literature suggests the inclusion of moderator variables in the models to improve prediction capacity and deal with the heterogeneity of the correlations considered in the meta-analysis [103].

##### Gender

Despite various inconclusive and diverging results, the literature emphasizes the importance of including the moderation of the gender variable in the new technology acceptance models [69,104,105]. The moderating variable of gender was encoded from the proportion of male respondents relative to the total (number of male respondents/total respondents).

##### Age Range

The literature reports a lower acceptance of e-health technologies by senior users than people in other age ranges [30,77]. In the studies considered in this meta-analysis, the demographic data referring to the age of the respondents is reported by grouping the ages in age ranges. Given this restriction, the age range variable corresponding to each study was estimated in this meta-analysis from the mean value between the limits of each age range considered in the studies weighted by the sample percentage corresponding to each studied range. When the age range’s maximum and minimum age limits were not defined (e.g., over sixty years old), 18 years was considered the minimum age, and 85 was the maximum age.

##### Timeline

The literature on meta-analysis suggests including the study year [40] as a moderator variable. Upon analyzing users’ acceptance of new technologies, the relations among its variables may change over time. A better understanding of this change over time still needs to be addressed in the literature [106]. For this reason, the publication year is considered a moderating variable in this study.

For the meta-analysis, the moderating variable “*Timeline*” was encoded as follows (Equation (1)), where *Year* = publication year of the analyzed study; *Year_Max_* = the most recent publication year among the studies considered for the analyzed relations; *Year_Min_* = the oldest publication year among the studies considered for the analyzed relations:(1)$$Timeline=\frac{\left(Year-Year_{Min}\right)}{\left(Year_{Max}-Year_{Min}\right)}$$

From the relations presented in the previous topics, the model proposed for this meta-analysis is represented in Figure 1. Hence, besides the relationships proposed by the UTAUT2 model (PE > BI, EE > BI, SI > BI, HM > BI, HB > BI, FC > BI, FC > UB) and the moderations considered in each relation (age range, gender, timeline), five relations of constructs considered important to the studied problem were included, primarily for presenting themselves frequently and with relevant results in the e-health literature (EE > PE, PE > AT, EE > AT, AT > BI, RP > BI).

### 3.2. Sample

The meta-analysis depends on the primary data. Thus, the execution of a comprehensive and quality bibliographic search becomes essential [107]. The methodological approach of Preferred Reporting Items for Systematic Reviews and Meta-Analyses (PRISMA—keywords protocol, eligibility criteria, information source, search and selection of studies) was used to elaborate the study. The search was carried out in the literature through the scientific databases “*Scopus*”, “*Web of Science*”, “*Emerald Insight*”, “*IEEE Xplore*” “*Science Direct*”, “*PubMed*”, and “*MedLine*”. The initial search on m-health technology acceptance articles [108,109,110] and technology acceptance models articles [36,111] raised the keywords used for the search. The keywords were organized into four search fields. The first search field corresponds to the analyzed technologies, “m-health” and “wearables,” applied with health purposes, combined with the words “health” and “fitness”. The second field refers to the user’s acceptance of new technology. The second field uses the combination of the words “accept*,” “engag*”, and “user”, with the asterisk indicating the inclusiveness of similar terms that have the same root. The third search field refers to the technology acceptance models. According to Taherdoost et al. [58], the most popular technology acceptance models are the following: “Technology Acceptance Model”, “Decomposed Theory of Planned Behavior” (DTPB), “Theory of Planned Behavior” (TPB), “Model of PC Utilization” (MPCU), “Theory of Reasoned Action” (TRA), “Innovation Diffusion Theory” (IDT), “Motivational Model” (MM), “Social Cognitive Theory” (SCT), “Unified Theory of Acceptance and Use of Technology” (UTAUT), and Unified Theory of Acceptance and Use of Technology 2”. The fourth search field corresponds to the statistical methods for estimating the models considered for the meta-analysis: “Partial Least Squares” and “Structural Equation Modeling.” The search used the topic procedure, restricting the search to the article’s field of title, abstract, and keywords. Table 1 displays the search string used in the databases. The search resulted in 273 studies. The details of the filtering mechanism of the materials selected for the meta-analysis are shown in Figure 2. The studies were filtered firstly based on the publication language. Next, 31 duplicated and 15 unavailable (articles with limited access to their full content) studies were excluded, leaving 238 papers to be analyzed fully.

### 3.3. Coding

The methodology approach used coding rules to guarantee consistency among the studies considered for this meta-analysis. As suggested by the literature [112], the initial pool of articles was assessed following criteria: empirical study containing at least one construct of the UTAUT2 model [113] or a similar one [36]; (ii) the presence of correlations among the constructs (*r_o_*); (iii) the internal consistency of the constructs (*r_xx_* and *r_yy_*); and (iv) the size of the sample utilized (*N_i_*). From the initial pool of 238 articles, the coding procedure found 84 articles that met all requirements. These 84 articles presented a total of 376 correlations among the constructs incorporated into the proposed model (Figure 1) and are based on a sample of 31,609 respondents (Appendix A). The number of articles, correlations, and the sum of respondents utilized in the current meta-analysis is relevant compared to other meta-analyses in the health area [114,115,116,117].

### 3.4. Analysis

The Hunter–Schmidt method [118] has been widely applied in studies that relate items measured by Likert scales and latent variables [119,120,121]. In this research, the correlations stemming from the studies considered for the meta-analysis (*r_o_*) were unattenuated using the reliability of the constructs for each relation as suggested by Hunter and Schmidt [47] (Equation (2)), where *r_m_* is the size of the effect corrected for the measurement error, and *r_xx_* and *r_yy_* are the reliabilities of the constructs involved in the relations, stemming from Cronbach’s alpha [122] or the Composite Reliability (CR) [123]:(2)$$r_{m}=\frac{r_{o}}{\sqrt{r_{xx}}\sqrt{r_{yy}}}$$

The correlations were corrected for the sampling error using the sample size of each observation as the weight (Equation (3)), where *r_c_* is the average corrected correlation for the bivariate relations, and *N_i_* and *r_i_* are the sample size and size of the effect corrected for the measurement error for each sample *i*, respectively:(3)$${\bar{r}}_{c}=\frac{\sum N_{i}r_{mi}}{\sum N_{i}}$$

Moreover, it is also possible to calculate the sampling error variance (*e_i_*) for each study (Equation (4)):(4)$$e_{i}=\frac{{\left(1-{\bar{r}}_{c}^{2}\right)}^{2}}{(N_{i}-1)r_{xx}r_{yy}{}^{2}}$$

The results of the relation estimates (${\bar{r}}_{c}$) and estimation errors (*e_i_*) were used to calculate the compiled effect of the relations. The individual estimates were treated as random effects, assuming that the correlations among the studies are different [40].

The method-denominated meta-regression was used to verify the need to incorporate moderators into the relations. Meta-regression is indicated as a way to analyze the heterogeneity of the residuals of the estimates through moderating variables [40,118]. To analyze the heterogeneity of the residues, the Q*_residuals_* statistic, which corresponds to a weighted measure of the square of errors, and the inconsistency test I^2^, which represents the proportion of studies in which the proposed model does not explain the coefficient, were considered [124].

From the techniques developed by Hunter and Schmidt [125] and presented by Borenstein et al. [118] and Card [40], we intended to identify the correlations among the constructs proposed in the model to measure the acceptance of e-health technologies by users. Statistics software Stata^®^ v. 16 was used to estimate all effects presented in this paper.

## 4. Results

### 4.1. Overview of the Studies Considered for the Meta-Analysis

Appendix B shows a growing trend of publications on the subject from the 84 studies selected and presented in the present meta-analysis. For example, this fact can be observed in the growth in the number of articles published over the years. It is also possible to observe that most of the studies considered for the meta-analysis come from China (17), followed by Bangladesh (9), the USA (7), and Taiwan (7). There is also a need for studies from Latin American countries. This fact may indicate the need to develop future studies to understand Latin American consumers’ acceptance of m-health technologies.

### 4.2. Reliability of Constructs

Table 2 presents the descriptive analysis of Cronbach’s alpha of the constructs considered in the meta-analysis. The results display that Cronbach’s alpha from all constructs is above 0.6, indicating that the constructs used in this meta-analysis are reliable [126].

### 4.3. Meta-Analysis of Model Correlations

The results of the estimates proposed in the model are presented in Table 3, indicating a significant effect for all relations proposed in the model. It is possible to observe that the AT > BI relation presented the most significant effect (β = 0.647; *p*-value < 0.05) among the considered relations. It is also possible to observe a negative effect resulting from the PR > BI relation. These results indicate that while Attitude has a more significant impact among users on their Behavioral Intention to Use an e-health device, the risk to privacy may cause resistance to this same intention to use.

It is also possible to observe that the HM > BI relation presented the smallest range in its Confidence Interval (β = 0.003; *p*-value < 0.05) and, consequently, a lower I^2^ value. These results indicate a smaller resulting variance among the effects corresponding to the HM > BI relation. However, the other values obtained for I^2^ point to high heterogeneity in the other relations considered for the meta-analysis, indicating the need to incorporate other moderating variables into the proposed model.

Regarding the moderating variables (Table 4), it is possible to verify that the effect of the moderating variables was significant for most of the relations proposed. Among the moderating effects that presented significant moderation (*p*-value < 0.05), it is possible to observe that the moderating effect of the “Timeline” in the FC > UB relation presented the highest coefficient (β = 1.2735; *p*-value = 0.026). The PE > BI, PV > BI, and BI > UB relations did not present a significant moderating effect by the variables considered in this study. It is also important to emphasize the high values obtained for I^2^ (except for the HM > BI relations), which indicate that even with the incorporation of three moderators, as suggested by the results in Table 3, the incorporation of other moderating variables is still necessary to better understand and estimate the relations.

## 5. Discussion

Figure 3 presents the results referring to the effects of the model proposed by the current meta-analysis, presenting the results for the coefficients previously shown in Table 3 and Table 4.

### 5.1. Main Relations of the Model

The results indicate a positive and significant relation (β = 0.339; *p*-value < 0.01) for the PE > BI relation. Although studies are verified indicating that the effect of the PE predictor is the most significant among the other constructs toward the BI [127,128], the effect of PE was the third-largest relative to the BI in this analysis. As a strategy to improve user understanding of the potential utility of e-health technologies, marketing professionals must communicate clearly the effectiveness of using the technology for health [1]. This indication is based on the positive perception that benefits stemming from using technology reinforce the intention to use a product [1].

For the EE > BI relation, the results indicate a positive and significant effect (β = 0.2320; *p*-value < 0.05). Although some studies indicate no significance for the relation EE > BI [129,130], the positive effect has been reported in many estimations in the literature [32,131,132]. The positive effect of the EE > BI relation is related to offering functions that meet user needs, promoting the increase in the acceptance of the effort required for use [133]. If the consumers perceive that using the technological device is intuitive and easy, they will more easily perceive the benefits and value of this technology [134]. As an alternative for those who are not acquainted with the used technologies, it would be possible to promote the reduction in the effort required to use the technology from the incorporation of graphical resources that allow the user greater facility to become familiarized with the available functionalities [135].

Users perceive greater utility in e-health devices (m-health/wearables) when they observe more ease in using the technology [99]. The results indicate a positive and significant effect (β = 0.4680; *p*-value < 0.01) for the EE > PE relation. The positive value for the coefficient indicates that the easy operation of e-health devices induces an increase in user expectations related to the desired performance for the technology to be acquired [69].

The meta-analysis estimated a significant (*p*-value < 0.01) and positive effect for the EE > AT and PE > AT relations (β = 0.3490 and β = 0.5250, respectively). Therefore, it is also important to highlight the effect of the AT construct on BI, which presented the most magnitude estimated in this meta-analysis (β = 0.6470; *p*-value < 0.01). In summary, these results indicate that users have a more positive attitude relative to e-health devices if the technology is perceived as useful [136] and easy to use [100]. The results also pointed out that a positive attitude directly influences Behavioral Intention.

Although some studies reported the non-significance of the relation SI > BI [130,137,138] and others still suggest a negative effect [139], the results of the meta-analysis indicate a positive and significant relationship (β = 0.2800; *p*-valor < 0.01) between SI > BI. The result may be explained by people’s desire to share views and behaviors perceived in specific groups [140]. The usefulness and reliability of the content were two criteria that predicted the participants’ intention to share digital information media [141]. The spread of misinformation on social media still causes fear and distrust among technology users [78]. (When the users indicate their acceptance of the technology before the community, the perception of risks tends to decrease, promoting more confidence in using the technological product [142]. In this sense, social networks are an important tool for forming opinions regarding products and brands due to the wide dissemination of information [32]. A better understanding of health outcomes from online information sharing becomes important for healthcare-related prevention and optimization [143]. Hence, the investment in resources directed at support and data collection from social media becomes essential.

For the relation FC > BI, the meta-analysis indicates a positive and significant trend (β = 0.4880; *p*-value < 0.01), although some studies indicate non-significance for this relation [30,33]. This result reflects the need for resources that improve internet services and increase compatibility among intelligent devices with health-monitoring functionalities [144]. The FC construct presents the fourth-largest effect among all 56 relations investigated in this meta-analysis. However, the literature considers that this construct is deemed one of the most important to determine BI due to the dependence of wearable and m-health devices on wireless network support and internet providers with high data transfer capacity [145].

The meta-analysis indicates a positive and significant effect (β = 0.2790; *p*-value < 0.01) for the relation FC > UB, although some studies reported non-significance for the same relation [77,146]. The positive effect of the FC > UB relation results from the positive influence of the presence of training and/or technical support capable of helping the user overcome concerns with technological innovations [145]. The presence of an operational structure capable of guiding the user simply or of a support system to obtain help positively influences the adoption of e-health technologies [32]. Training programs, technical support, and financial aid provided by professionals or family members would be crucial for using e-health devices [145]. The updates to enhance e-health product functionalities may even occur through continuous improvement, employing big data analyses related to medical care [139].

Several studies suggest that Hedonic Motivation plays a direct role in the Behavioral Intention to Use e-health technologies [30,147]. The meta-analysis indicates a positive and significant effect (β = 0.1150; *p*-value < 0.01) for the HM > BI relationship. This positive effect may indicate that the studied devices improve social communication and pleasure in using these technologies, besides the use purpose related to health monitoring [147].

Comparing the estimations obtained in studies aimed at m-health acceptance by teenagers [129] and elderlies [148], it appears that the HB > BI relationship is non-significant for the first case and significant for the second. Such divergence suggests that age may be a relevant moderating factor to be considered. Even with these divergences, the meta-analysis indicates a positive and significant coefficient (β = 0.364; *p*-value < 0.10) for the relation HB > BI. A positive effect of this relationship indicates that adopting a permanent habit increases the likeliness of accepting the studied technologies [36]. These results may also represent the user’s dependence on the habitual use of such devices [130].

For the relation PV > BI, although some studies in the literature on the subject suggest the significance of this relation [149,150], the meta-analysis estimated a non-significant relationship (*p*-value > 0.05). The acceptance of a given technology tends to increase insofar as the user perceives that the benefits of using such technologies are superior to the cost of their adoption [132,151]. The non-significant coefficient for the PV > BI relationship may be related to the great variety of e-health devices available and the benefit provided by these devices [144].

Among the studies considered for the present meta-analysis, only one showed a positive sign for the PR > BI relationship [152]. Intuitively, it is reasonable to expect that PR has a negative effect since this construct represents a consumer concern. Nonetheless, the authors justified the result because older adults are less concerned about privacy, and PR would not hinder older people’s acceptance of the e-health device [152]. Despite that, the meta-analysis estimated a negative and significant coefficient for the relationship between PR > BI (β = −0.1600; *p*-value < 0.01). The literature points to the concern of patients with the possibility of disseminating personal health information [153]. M-health devices such as m-health and wearable devices are more vulnerable to attacks and information interception, contributing to user insecurity regarding the privacy of such devices [101]. The perception of privacy becomes even more important when disclosing personal health information that may cause embarrassment to the user [154,155,156]. From this, developers must make sure that e-health devices comply with the data collection, processing, and storage regulations and provide transparency to consumers regarding data collection and use [101]. In summary, managers must conduct product design plans aligned with marketing strategies and privacy protection policies to attract consumers [157].

The literature reported that behavioral intention does not always indicate the actual use of the technology [158,159]. However, many estimations indicated that the Usage Behavior (UB) of an e-health technology is preceded and strongly affected by Behavioral Intention (BI), [26,160,161,162]. The meta-analysis indicates a positive and significant relationship for BI > UB (β = 0.525; *p*-value < 0.01). Hence, Behavioral Intention (BI) is an efficient indicator of the actual Usage Behavior of users.

### 5.2. Relations of the Moderating Variables

Firstly, it is possible to observe that only three relationships were significantly moderated by the three moderating variables proposed for the model (EE > AT, AT > BI, and FC > UB). Secondly, it was verified that some relations were not significantly influenced by the moderators proposed in this work (PE > BI, EE > PE, PV > BI, BI > UB). However, as presented before, most relationships present I^2^ values close to 100% (except the HM > BI relations). The I^2^ values suggest that the inclusion of more moderating variables into the model is necessary to deal with the heterogeneity of the residuals. Hence, although the PE > BI, EE > PE, PV > BI, and BI > UB relations were not significantly influenced by the proposed moderators, the incorporation of other moderations could reveal significant effects on these relations, enabling an adjustment for the proposed model.

#### 5.2.1. Gender

Gender exerts an important effect on adopting e-health technologies [146], which may be observed in the meta-analysis results. The moderation of the gender variable is significant (*p*-value < 0.05) for six of the relationships (PE > AT, β = 0.5012; EE > BI, β = 0.7610; EE > AT, β = 1.0522; AT > BI, β = 0.4486; HM > BI, β = 0.0078; PR > BI, β = 1.0522). These significant moderating effects indicate a greater influence of these relationships in men than in women. The effect was more relevant in men for relations involving HM and AT, which can be explained by the fact that men are more adventurous and are more likely to explore new technologies. At the same time, women desire factors that give them security (support) for the use of a technological system [163]. Although women tend to be more attracted by mobile technologies [164], men are more inclined to adopt m-health systems [165].

Some authors have suggested that women use less technology [166] and are less acquainted with new technologies [167,168]. The literature also indicates that the EE > BI relationship may influence men more to accept e-health devices [28,146], agreeing with the results obtained in the current meta-analysis.

#### 5.2.2. Age Range

User age range affects adopting e-health technologies [98]. The current meta-analysis results indicate that the age range’s moderation was significant for five of the relations (*p*-value < 0.05). Most of the significant effects of the “age range” moderator resulted in positive coefficients (EE > AT, β = 0.5799; SI > BI, β = 0.3163; AT > BI, β = 0.5863; HB > BI, β = 0.8443), which point out that older people are more susceptible than younger people to these relationships. Previous studies on adopting new technologies have suggested that the perceived benefits of technology influence the intention of senior citizens to adopt the technology [30,146,169]. It is verified in the literature that older people tend to be more susceptible to the complexity of technology [170]. Older adults with relatively less experience with the internet find a more challenging environment searching for reliable information [171]. Despite e-health motivating the elderly with health care, this motivation can be reduced over time due to the perception of the incompatibility of these technologies with the social environment of the elderly [172]. However, senior citizens have a positive attitude toward adopting technologies that render their lives more convenient [173] and make them more independent and with better quality of life [174]. The attitude most strongly linked to the perception of older people relative to the Behavioral Intention to Use technology may be related to health problems and concerns, which tend to increase with age [175]. With the increase in age, it is also possible to perceive that ease of use is considered a relevant factor for the attitudes of users toward the adoption of the technology (EE > AT) [100]. As the older population acquires a more relevant proportion relative to the general population, understanding their specific needs is essential to increase their technological acceptance level [98,131]. The FC > UB relation for this analysis was the only one significant for younger people (FC > UB, β = −0.8599). The development of mobile devices influences more and more youths to monitor their health and have healthier lifestyles continuously [176].

#### 5.2.3. Timeline

From the results obtained, it is possible to observe the significance of the moderating variable of timeline. The meta-analysis showed that the moderation presented a negative sign in five relationships (PE > AT, β = −0.4415; EE > AT, β = −0.5324; AT > BI, β = −0.3046; SI > BI, β = −0.3046; FC > BI, β = −0.1898; HB > BI, β = −0.1767). These results indicate that the magnitude of estimated relationships has decreased over time. The resulting values may be explained by the fact that people are more used to a technological environment, which would enable more considerable reliability of the performance of the technology, less concern regarding the effort required for its use, and more regularity in using technological devices.

In contrast, the FC > UB and PR > BI relationships presented positive values (FC > UB, β = 1.2735; PR > BI, β = 0.3693), which indicates that the intensity of the effects of FC on UB and PR on BI has increased over the years. The result corresponding to the FC > UB relation may be translated through the greater need for a structure that serves health requirements and allows speed in the transport of information and technological ubiquity. In turn, the result referring to the PR > BI relation may indicate the increase in the consumers’ concern with the security of confidential information, corresponding to the vulnerability of mobile digital services stemming from the high rate of information transfer among networks [101].

### 5.3. Implications for Theory and Practice

As the main contribution, this study presents a general guide to understanding how the process of accepting new e-health technologies takes place, indicating overall guidance for future research and the development of such technologies considering their acceptance by users. In the academic context, the results of this meta-analysis present significant variables (e.g., PE > AT, AT > BI, BI > UB) that must serve as guidelines for future research on the acceptance of other e-health technologies. Furthermore, the non-significance of some of the relations among the analyzed constructs (e.g., PV > BI and some moderations of the relationships) suggests that future investigations should explore such relations considering incremental alterations to the proposed model.

As for the practical implications, the results guide the marketing and product development activities of m-health and wearable devices. Managers and developers can obtain direction for their activities from the degree of importance of each construct analyzed. It is necessary to consider the individuality of each user to provide more flexible solutions with greater capacity for the customization of health information-sharing services [177]. Understanding consumers’ needs enables focusing on developing components essential to the market acceptance of the studied devices.

### 5.4. Limitations and Directions for Future Research

Calculating the I^2^ made it possible to verify the need to include other moderating variables to improve the model fit. The results indicate that these relationships still present significant heterogeneity, evincing that other factors still not considered by this study also affect the acceptance by users of e-health technologies. Hence, future research may explore the model proposed in this study, adding other moderating variables that may help explain the studied phenomenon.

There is a concern for the World Health Organization in reaching equity in health services for medical issues and social matters. Hence, similar studies must be able to analyze possible users residing in underdeveloped or developing countries, as is the case for countries located in Latin America, especially Brazil. This fact becomes relevant due to the variation in the cultural and economic characteristics of the population and characteristics associated with the regulation and structuring of health services that may influence technology acceptance. Moreover, future research must assess which components are considered essential for these devices from the perception of general and specific users.

## 6. Conclusions

This study sought to understand consumers’ acceptability relative to e-health technologies associated with m-health and wearable devices. A meta-analysis was carried out considering 84 previous studies, a total of 31,609 respondents, and 376 correlations.

Through the meta-analysis, fourteen relations among the constructs were estimated as significant. The effect of the AT construct on BI (AT > BI) presented the highest intensity (β = 0.6470; *p*-value = 0.00). The model is also composed of 23 effects of moderating variables on the relationships. Only the PE > BI, EE > PE, PV > PC, and BI > UB relations were not significantly affected by any proposed moderating variables.

## Figures and Tables

**Figure 1 ijerph-20-04369-f001:**
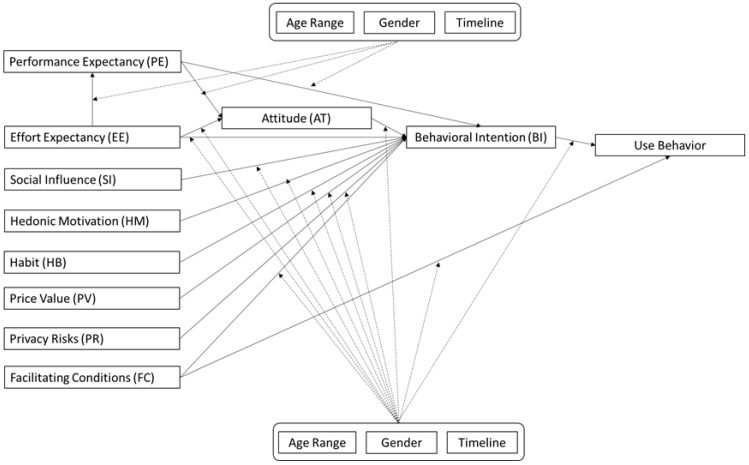
Proposition of the model based on the UTAUT2 used in the meta-analysis.

**Figure 2 ijerph-20-04369-f002:**
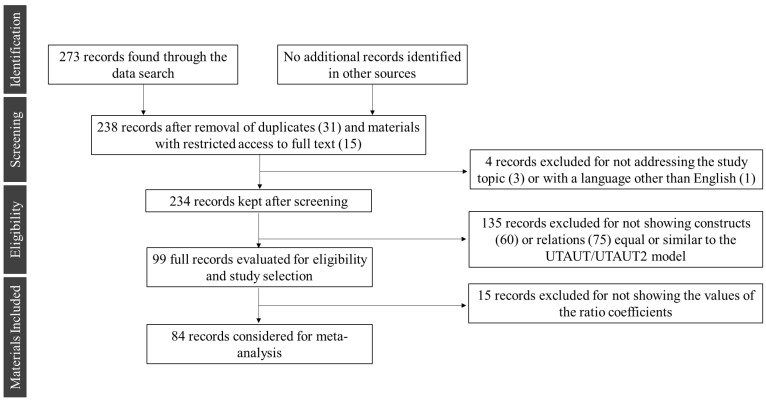
Details of the filtering mechanism of the materials selected for the meta-analysis.

**Figure 3 ijerph-20-04369-f003:**
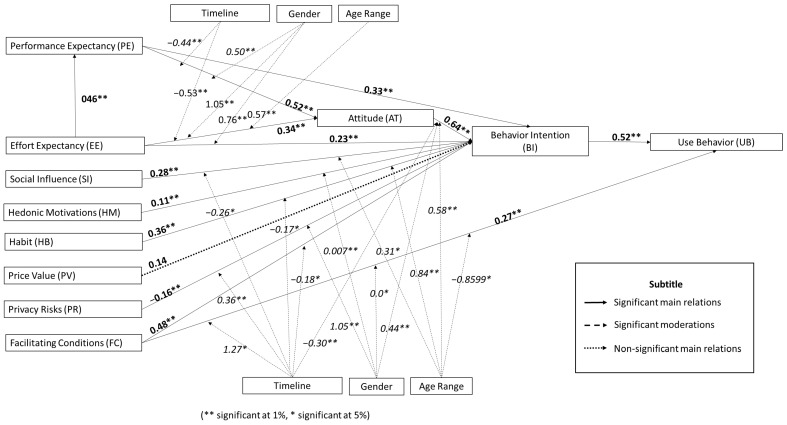
Meta-analysis model for e-health technology acceptance, corresponding to m-health and wearable devices.

**Table 1 ijerph-20-04369-t001:** Search string used in the databases for the search mechanism.

Search Field	Technologies		Acceptance of New Technology		Models of Technolgy Acceptance		Statistical Methods
Search keywords	(m-health AND Health) OR (m-health AND Fitness) OR (Wearable AND Health) OR (Wearable AND Fitness)	AND	(User AND Accept*) OR (User AND Engag*)	AND	Technology Acceptance Model OR TAM OR Decomposed Theory of Planned Behavior OR DTPB OR Theory of Planned Behavior OR TPB OR Model of PC Utilization OR MPCU OR Theory of Reasoned Action OR TRA OR Innovation Diffusion Theory OR Motivational Model OR MM OR Social Cognitive Theory OR SCT OR Unified Theory of Acceptance and Use of Technology OR UTAUT OR Unified Theory of Acceptance and Use of Technology 2 OR UTAUT2	AND	Partial Least Squares OR Structural Equation Modeling

**Table 2 ijerph-20-04369-t002:** Descriptive statistics of the Cronbach’s alpha values stemming from the considered constructs.

	EP	EE	AT	IS	CF	MH	HB	VP	RP	IC	CU
Average reliability	0.830	0.841	0.787	0.848	0.821	0.876	0.859	0.831	0.878	0.852	0.855
Minimun	0.657	0.650	0.700	0.642	0.690	0.779	0.649	0.700	0.700	0.652	0.700
Maximum	0.978	0.970	0.968	0.970	0.945	0.976	0.973	0.940	0.952	0.976	0.972
Number of samples	74	71	21	44	35	21	9	11	8	76	9

**Table 3 ijerph-20-04369-t003:** Effects corresponding to the main variables of the meta-analysis.

Relations	N (Total)	K (Estudies)	Coefficient	Confidence Interval (95%)	Estat. Θ	*p*-Value
PE > BI	26,098	77	0.339	[0.293; 0.385]	14.30	<0.01 **
PE > AT	7030	17	0.525	[0.475; 0.575]	20.54	<0.01 **
EE > PE	9028	21	0.468	[0.389; 0.548]	11.53	<0.01 **
EE > BI	21,358	67	0.232	[0.188; 0.277]	10.22	<0.01 **
EE > AT	5901	16	0.349	[0.270; 0.428]	8.640	<0.01 **
AT > BI	2093	10	0.647	[0.510; 0.784]	9.250	<0.01 **
SI > BI	18,600	57	0.280	[0.224; 0.337]	9.740	<0.01 **
FC > BI	14,492	43	0.164	[0.118; 0.211]	6.940	<0.01 **
FC > UB	1812	5	0.279	[0.124; 0.434]	3.530	<0.01 **
HM > BI	6531	24	0.115	[0.113; 0.116]	147.8	<0.01 **
HB > BI	2526	9	0.364	[0.236; 0.492]	5.580	<0.01 **
PV > BI	3462	12	0.148	[−0.040; 0.336]	1.540	0.12
PR > BI	3519	8	−0.160	[−0.256; −0.070]	−3.340	<0.01 **
BI > UB	5438	10	0.488	[0.358; 0.618]	7.350	<0.01 **

** significant at 1%.

**Table 4 ijerph-20-04369-t004:** Meta-regression of the moderating variables.

Moderator	Coef	Std. Error	Z	*p* > |z|	I^2^ (%)
PE > BI					
Gender	−0.20	0.21	−0.97	0.33	99.97
Age Range	0.16	0.11	1.48	0.13	
Timeline	−0.09	0.10	−0.87	0.38	
PE > AT					
Gender	0.50	0.16	2.98	<0.01 **	99.93
Age Range	0.13	0.08	1.53	0.12	
Timeline	−0.44	0.10	−4.06	<0.01 **	
EE > PE					
Gender	−0.06	0.27	−0.25	0.80	99.86
Age Range	0.12	0.10	0.11	0.91	
Timeline	−0.24	0.14	−1.7	0.08	
EE > BI					
Gender	0.76	0.21	3.50	<0.01 **	99.97
Age Range	0.14	0.11	1.27	0.20	
Timeline	0.05	0.10	0.53	0.59	
EE > AT					
Gender	1.05	0.29	3.61	<0.01 **	99.89
Age Range	0.57	0.12	4.52	<0.01 **	
Timeline	−0.53	0.19	−2.78	<0.01 **	
AT > BI					
Gender	0.44	0.05	8.00	<0.01 **	88.21
Age Range	0.58	0.02	24.32	<0.01 **	
Timeline	−0.30	0.03	−7.64	<0.01 **	
SI > BI					
Gender	−0.16	0.22	−0.73	0.46	99.98
Age Range	0.31	0.15	1.99	0.04 *	
Timeline	−0.26	0.11	−2.25	0.02 *	
FC > BI					
Gender	−0.31	0.18	−1.68	0.09	99.95
Age Range	0.13	0.10	1.22	0.22	
Timeline	−0.18	0.08	−2.23	0.02 *	
FC > UB					
Gender	0	-	−2.22	0.02 *	99.98
Age Range	−0.85	0.38	2.09	0.03 *	
Timeline	1.27	0.61	2.42	0.01 *	
HM > BI					
Gender	<0.01	<0.01	0.97	<0.01 **	3.37
Age Range	0.06	<0.01	19.68	0.06	
Timeline	0.16	<0.01	61.97	0.16	
HB > BI					
Gender	0.01	0.22	0.08	0.93	99.49
Age Range	0.84	0.10	8.06	<0.01 **	
Timeline	−0.17	0.07	−2.22	0.02 *	
PV > BI					
Gender	0.15	0.23	0.69	0.49	99.56
Age Range	−0.07	0.20	−0.36	0.71	
Timeline	0.01	0.19	0.06	0.95	
PR > BI					
Gender	1.05	0.35	2.93	<0.01 **	99.68
Age Range	0.03	0.15	0.24	0.80	
Timeline	0.36	0.12	2.85	<0.01 **	
BI > UB					
Gender	0.27	0.69	0.39	0.69	99.99
Age Range	0.18	0.23	0.77	0.44	
Timeline	0.23	0.22	1.04	0.29	

** significant at 1%, * significant at 5%.

## Data Availability

The data presented in this study are available on request from the corresponding author.

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
