# Peer review of "Getting Connected to M-Health Technologies through a Meta-Analysis"

_ijerph, 2023, doi:10.3390/ijerph20054369_

Round 1
Reviewer 1 Report
The article needs improvements regarding clarity and completeness. for instance:
- I recommend the following modification in the title of the article: Getting Connected to M-Health Technologies through an Meta-Analysis --> Getting Connected to M-Health Technologies through a Meta-Analysis.
- The authors state "Demand for mobile e-health technologies (m-health) continues in constant growth, stim-9 ulating the technological advance of such devices. ". Are m-health referring to applications, applications/devices, or devices? The sentence seems to only relate m-health to devices. The next sentence confirms such an inference. Is it correct?
- The authors state "their users' acceptance of the studied e-health systems.". Is it e-health or m-health?
- technologies applied to health --> technologies applied to healthcare
- health care --> healthcare
- of people with health --> of people with their health
- advances in IoT devices --> advances in Internet of Things (IoT) devices
- I recommend the authors, after defining the e-health and m-health terms, use the m-health term throungout the docuent.
- I recommend the authors include a related works section to discuss existing studies and limitations. For instance, if there is no meta-analysis on this topic, the authors could discuss systematic reviews and point out such a limitation.
- The acronym IoT should be defined in the first mention.
- Theory of Acceptance and Use of Technology (UTAUT) shoud be defied at the first mention.
- Figure 1 should be explained after citation. The conections between the elements of Figure 1 are hard to understand.
- IEEE --> IEEE Xplore
- The authors state "According to Taherdoost et al. (Taherdoost 2018), ". Please use the correct citation format.
- Technology Acceptance 207 Model (TAM), and others, are defined twice in the article. Please revise the formating of the text in Page 5 of the PDF file (Section 4.1). The division of paragraphs seems to be incorrect.
- Please provide a figure to summarize the search results. PRISMA provides an exemple of syntesis using a figure.
- The revision protocol description needs to be further detailed. For instance, the authors should present a more clear description of inclusion and exclusion criteria. It is also relevant to present data extraction fields. Do the authors evaluate the quality of papers?
- Besides, could the authors present syntesize data collected from the papers using some figures? In my opinion, such synthesis would introduce readers relevant information on the selected papers (e.g., publication year, focus, and experiment data).
- Could the authors provide a table with basic information (e.g., titles) of the 83 papers?
- Figure 2 should be further explained after citation.
- Is the following text part of Figure 2? ** significant at 1%, * significant at 5% (above the title of Section 6.1.). This formating is confusing.
- In this context, --> Therefore,
Such improvements can improve readability and enable a more detailed evaluation of results.
Author Response
Thank you very much for the collaboration. Your comments were constructive overall, and we appreciated this constructive feedback. After addressing the issues raised, the quality of the paper has improved, and we hope you agree.
Best regards,
The authors.

Reviewer 2 Report
Title: Getting Connected to M-Health Technologies through an Meta-2 Analysis
Demand for mobile e-health technologies (m-health) continues in constant growth, stimulating the technological advance of such devices. However, the customer needs to perceive the utility of these devices to incorporate them into their daily lives.
The author aim to synthesize the literature results related to the customer's acceptance of m-health technologies.
They used the relations and constructs proposed in the UTAUT2 technology acceptance model. Their methodological approach utilized a meta-analysis to raise the effect of the main factors on Behaviour Intention of Use m-health technologies. Furthermore, the model proposed also estimated the moderation effect of gender, age, and timeline variables on the UTAUT2 relations.
Their meta-analysis utilized 83 different articles, which presented 376 estimations based on a sample of 31,609 respondents. Results indicate an overall compilation of the relations, as well as the primary factors and moderating variables that determine their users' acceptance of the studied e-health systems.
The study is interesting.
Some improvements are needed.
Strengths:
The work is exhaustive.
The proposed model is interesting.
Points of weakness:
It would be necessary to better illustrate the methodology that leads to the creation of the model which in the present draft seems mixed between introduction and methods.
The use of follow charts must be supported by an accurate description.
Further comments:
1. The abstract must be rewritten better summarizing the sections.
2. The aims are not clear “This study aims to synthesize the literature estimations on the acceptance of m-health technologies.” Explain better the key questions. Use bullet points. Insert the definition of the term acceptance.
3. Section 2-3 and 4 must be revised. Methods and introductive aspects are mixed.
4. Describe Figure 1 in details (this must be applied to all figures).
5. Formulas must follow the MDPI standard and numbered in the body of the manuscript.
6. The themes proposed in the results must be introduced by means of a few sentences.
7. Figure 2 is terribly proposed. Improve it and comment in details.
8. Move the limitations in the discussion.
Author Response

(The authors gave the same response as above.)

Round 2
Reviewer 1 Report
The article still needs some improvements, and some of my requests remain unaddressed. For instance:
- Please define the UTAUT2 acronym (first usage) in the abstract.
- Could Section 2 be renamed to Acceptance Models? It seems discussing such models is the main focus of the section.
- In my opinion, it is not necessary to define acronyms twice. For instance Internet of Things (ioT).
- The authors state: Hence, besides the relations proposed by the UTAUT2 model, five relations of constructs considered critical to the acceptance of m-health technologies were included. The relation between Effort Expectancy (EE) > Performance Expectancy (PE) constructs, as suggested by the literature [91,92]. The relation between Performance Expectancy (PE) > Attitude (AT) [93,94]. The literature also suggests estimating the relation between Effort Expectancy (EE) > Attitude (AT) [95,96], Attitude 138 (AT) > Behavioral Intention (BI) [97,98], and Privacy Risks (RP) > Behavioral Intention (BI) [91,99].
Could you itemize it? For instance: Hence, besides the relations proposed by the UTAUT2 model, five relations of constructs considered critical to the acceptance of m-health technologies were included:
\begin{itemize}
\item The relation between Effort Expectancy (EE) > Performance Expectancy (PE) constructs, as suggested by the literature [91,92].
\item The relation between Performance Expectancy (PE) > Attitude (AT) [93,94].
\item The literature also suggests estimating the relation between Effort Expectancy (EE) > Attitude (AT) [95,96], Attitude (AT) > Behavioral Intention (BI) [97,98], and Privacy Risks (RP) > Behavioral Intention (BI) [91,99].
\end{itemize}
- precious topics --> previous topics?
- figure 2 --> Figure 2
- The authors stated that "15 unavailable studies were excluded ". Please clarify what unavailable means for your study. What type of restricted access?
- Details of the filtering mechanism of the materials selected for the meta-analysis --> Details of the filtering mechanism of the materials selected for the meta-analysis.
- In my opinion, one more subsection in Section 4 (a new first one) with the aim to present an overview of the 83 accepted articles. For instance, the authors could use a figure to synthesize the years of publication (i.e., the timeline). This is only an example of a data extraction field the authors could use to present the synthesis. I recommend the authors focus on the main data extraction field to present such a synthesis (e.g., constructs). Presenting this overview would increase the clarity and readability of the article.
Author Response

(The authors gave the same response as above.)
